# Safety, Efficacy, and Immunogenicity of Therapeutic Vaccines for Patients with High-Grade Cervical Intraepithelial Neoplasia (CIN 2/3) Associated with Human Papillomavirus: A Systematic Review

**DOI:** 10.3390/cancers16030672

**Published:** 2024-02-05

**Authors:** Caroline Amélia Gonçalves, Gabriela Pereira-da-Silva, Renata Cristina Campos Pereira Silveira, Paulo César Morales Mayer, Adriana Zilly, Luís Carlos Lopes-Júnior

**Affiliations:** 1Maternal-Infant and Public Health Nursing Department, University of São Paulo at Ribeirão Preto School of Nursing, Campus Ribeirão Preto, Ribeirão Preto 14040-902, Brazil; coord-biomedicina.itz@ceuma.br (C.A.G.);; 2Universidade CEUMA, São Luís 65075-120, Brazil; coord-psicologia.itz@ceuma.br; 3Center for Education, Literature and Health, State University of West of Parana, Cascavel 85819-110, Brazil; 4Health Sciences Center, Universidade Federal do Espirito Santo (UFES), Av. Marechal Campos, 1468—Maruípe, Vitoria 29043-900, Brazil

**Keywords:** neoplasms, HPV, vaccines, immunogenicity of the vaccine

## Abstract

**Simple Summary:**

We conducted a systematic review using 16 trials (RCTs and NRCTs) across eight countries and three continents; which included 672 patients with high-grade CIN associated with HPV. To the best of our knowledge, this is the first synthesis related to the safety, efficacy, and immunogenicity of therapeutic vaccines for the treatment of patients with high-grade CIN associated with HPV. The findings from the trials revealed that therapeutic vaccines can be considered promising for these injuries. However, there are still only a few phase III RCTs, and a better understanding of the specificities of different vaccine compositions and delivery systems and pathways to prevent viral escape mechanisms are needed. Overall, the results of this systematic review indicate that the therapeutic vaccines that are being developed for the treatment of CIN 2/3 are safe and well tolerated and that most trigger only systemic symptoms classified as mild or moderate that improve spontaneously in a short period of time. Considering the inconsistent results among phase I and II trials involving different therapeutic vaccines, our findings provide some clarification and have implications for multiple stakeholders.

**Abstract:**

Despite the knowledge that HPV is responsible for high-grade CIN and cervical cancer, little is known about the use of therapeutic vaccines as a treatment. We aimed to synthesize and critically evaluate the evidence from clinical trials on the safety, efficacy, and immunogenicity of therapeutic vaccines in the treatment of patients with high-grade CIN associated with HPV. A systematic review of clinical trials adhering to the PRISMA 2020 statement in MEDLINE/PubMed, Embase, CENTRAL Cochrane, Web of Science, Scopus, and LILACS was undertaken, with no data or language restrictions. Primary endpoints related to the safety, efficacy, and immunogenicity of these vaccines were assessed by reviewing the adverse/toxic effects associated with the therapeutic vaccine administration via histopathological regression of the lesion and/or regression of the lesion size and via viral clearance and through the immunological response of individuals who received treatment compared to those who did not or before and after receiving the vaccine, respectively. A total of 1184 studies were identified, and 16 met all the criteria. Overall, the therapeutic vaccines were heterogeneous regarding their formulation, dose, intervention protocol, and routes of administration, making a meta-analysis unfeasible. In most studies (n = 15), the vaccines were safe and well tolerated, with clinical efficacy regarding the lesions and histopathological regression or viral clearance. In addition, eleven studies showed favorable immunological responses against HPV, and seven studies showed a positive correlation between immunogenicity and the clinical response, indicating promising results that should be further investigated. In summary, therapeutic vaccines, although urgently needed to avoid progression of CIN 2/3 patients, still present sparse data, requiring greater investments in a well-designed phase III RCT.

## 1. Introduction

It is estimated that 80% of sexually active people will be affected by at least one human papilloma virus (HPV) type at some point in life [1,2], indicating a high prevalence of the virus, especially in underdeveloped countries. Although the presence of the virus does not imply cancer, it is a necessary condition for high-grade cervical intraepithelial neoplasia (CIN) and cervical cancer, constituting a major global public health problem [3,4], mainly due to the high morbidity and mortality of HPV-related diseases [5].

High-risk HPVs, especially types 16 and 18, are present in approximately 100% of cases of CIN 2 and 3 and may progress to several types of cancer, especially cervical cancer [6]. The treatments currently used for CIN may result in recurrent or persistent infections because of the incomplete elimination of the virus. Additionally, conventional therapies are associated with reproductive and psychological impairments, with negative impacts on the quality of life of patients [7,8,9]. Furthermore, a larger problem is also the lack of access to these vaccines in low-income countries, particularly since therapy for CIN is less widely available in low-income countries [10].

The prevention of neoplasms occurs through prophylactic vaccines, which, although safe and effective, cannot eliminate already established lesions and have no effect on already established lesions caused by HPV [10,11,12]. HPV uses aggressive immune evasion strategies via the expression of oncoproteins E6 and E7, which induce the hyperproliferation of keratinocytes, making the virus less liable to an immune attack. These oncoproteins are involved in the disruption of cell cycle checkpoints and the modulation of the host immune response, blocking gene expression in these cells and favoring an immunosuppressive environment. Additionally, it interferes in the activation of adaptive immune cells and in the release of pro-inflammatory cytokines. These mechanisms allow viral replication, promoting cancer development [13]. The products of oncogenic viruses cause specific T cell responses, in addition to responses by other cells of the innate and acquired immune systems; these responses function to eradicate viruses. Tumors induced by viruses are considered to be the most immunogenic because they originate from antigens that are foreign to our bodies [14].

Therapeutic vaccines aim to stimulate cell-mediated immune responses against specific antigens and promote the death of infected cells. They are used in cases where the disease is already established, originating from persistent or recurrent lesions, to promote the regression of precancerous lesions and the remission of invasive cancer [15,16,17]. Currently, there are several therapeutic vaccine candidates in preclinical studies as well as in clinical trials; however, there has been no demonstration of efficacy in phase 3 studies, which support the licensing of these vaccines yet.

Despite a growing body of literature pointing to the development of therapeutic vaccines for CIN, there is still no systematic review that synthesizes the state of the art of clinical trials simultaneously taking into account the three endpoints (safety, efficacy, and immunogenicity) of these therapeutic vaccines. Hence, the purpose of this study was to synthesize and critically evaluate the evidence from clinical trials on the safety, efficacy, and immunogenicity of therapeutic vaccines for the treatment of patients with CIN 2/3 associated with HPV.

## 2. Materials and Methods

### 2.1. Search Strategy and Selection Criteria

This systematic review adhered to the PRISMA 2020 statement [18]. In addition, this study was registered with International Prospective Register of Systematic Reviews (PROSPERO), under registration ID: CRD42017077428. Using the PICOS [19] strategy, our formulated research question was: “What is the scientific evidence from clinical trials on the safety, efficacy and immunogenicity of therapeutic vaccines administered to patients with high-grade cervical intraepithelial neoplasia associated with HPV?” (Table 1).

Six online bibliographic databases were searched from their date of inception to 20 August 2023: Medical Literature Analysis and Retrieval System Online—MEDLINE/PubMed (from 1946); Excerpta Medica Database—EMBASE (from 1946); Cochrane Central Register of Controlled Trials (from 1996); Web of Science (from 1900); Scopus (from 2004); and LILACS (from 1967). There was no date or language restriction in the search strategy. In addition to the aforementioned databases, secondary searches were performed in other sources, such as ClinicalTrials.gov (National Institutes of Health—NIH, Bethesda, MD, USA); Brazilian Clinical Trials Registry (ReBEC); The British Library; Scientific Electronic Library Online—SciELO; and Google Scholar. The references in the included studies were manually analyzed to find additional relevant studies. All the steps of this systematic review were independently performed by 2 researchers (CAG and LCLJ). The reference manager EndNote™ was used to store, organize, and exclude duplicates to ensure a systematic and manageable search. The database searches were conducted in November 2018 and updated in August 2023. Appendix A shows the complete search strategy for each database.

Primary studies (phase I, II, or III RCT or quasi-experimental studies, NRCT) conducted with patients with high-grade CIN 2/3 associated with HPV and no associated immunodeficiency were included. In addition, we included trials in which patients received therapeutic vaccines, regardless of the route of administration, and that evaluated the safety, efficacy, and immunogenicity endpoints through comparisons with a control group (placebo or standard treatment) or with each patient’s own parameters before and after vaccine administration (for NRCT). Studies that evaluated only 1 or 2 of the 3 proposed endpoints (safety, efficacy, and immunogenicity) or evaluated the vaccine in men were excluded.

The primary endpoints evaluated were the safety, efficacy, and immunogenicity of the therapeutic vaccines. Therefore, we assessed safety by analyzing the adverse and toxic effects associated with the administration of therapeutic vaccines. Efficacy was assessed via histopathological regression of the lesion and/or regression of the lesion size as well as via viral clearance. The immunogenicity of therapeutic vaccines was assessed by comparing the immunological adaptative response in serum or in peripheral blood mononuclear cells and target tissue of individuals who received treatment with that of those who did not or by comparing factors in individuals before and after receiving the vaccine.

Initially, the selection of articles was based on information contained in the title and abstract of each study and was independently performed by 2 researchers (CAG and LCLJ). Full-text reading of the articles was independently performed by the researchers after the initial selection. Cohen’s kappa coefficient was used to estimate the index of agreement between the 2 evaluators in each review phase (selection, extraction, and methodological evaluation of the included studies). Discrepancies were resolved through discussions at each stage, and a consensus was achieved, with acceptable inter-rater reliability (k = 0.93). A third researcher (RCCPS) verified the eligibility of the included studies.

### 2.2. Data Analysis

Two researchers (CAG and LCLJ) independently extracted the following data using pre-established and adapted tools [20,21,22,23]: (I) study characteristics (article title, country of origin of the study authors, year of publication, study host institution (hospital, university; research center, multicenter study, or study in a single institution), conflicts of interest, and funding); (II) methodological characteristics (study design, trial register, location, study objective or research question or hypotheses, sample characteristics, e.g., sample size, inclusion and exclusion criteria, ethical issues, baseline characteristics of the experimental and control groups, recruitment method, randomization, masking, intervention protocol, drop-outs, duration of follow-up, procedures for data collection, outcomes and statistical analysis); (III) main findings and implications for clinical practice; and (IV) limitations and conclusions (Appendix A). For data extraction, 2 Microsoft Excel^®^ (version 16.67 for Mac Book pro) spreadsheets were prepared by the researchers (CAG and LCLJ) to synthesize the data from the included studies. After this phase, the data were compiled into a single spreadsheet before proceeding with analyses. In addition, if data were missing or unclear or the nature of the intervention was unclear, we contacted the corresponding author of the publication via email for clarification.

The internal validity and risk of bias of RCT were assessed using the revised Cochrane Risk-Of-Bias tool for randomized trials (RoB 2) [24], which assesses the risk of bias in 5 domains: (1) randomization process; (2) deviations from the intended interventions; (3) missing outcome data; (4) measurement of the outcome; and (5) selection of the reported result [24]. The RoB 2 classifies risk of bias as follows: (1) low risk of bias: low risk of bias for all domains; (2) some concerns: some concerns in at least 1 domain, but no high risk of bias for any domain; and (3) high risk of bias: high risk of bias in at least 1 domain or some concerns for multiple domains, substantially reducing the confidence in the result [24]. To assess NRCTs, the Risk of Bias in Non-randomized Studies of Interventions (ROBINS-I) was used [25]. The ROBINS-I comprises 7 chronologically arranged bias domains (pre-intervention, at intervention, and post-intervention), and the domain level and overall risk of bias are classified as low, moderate, serious, or critical [25]. Using both tools (RoB2 and ROBINS-I), the same 2 reviewers (CAG and LCLJ) independently assessed the risk of bias for each included study. Discrepancies were resolved through a discussion at each stage, and a consensus was achieved, with acceptable inter-rater reliability (k = 0.93). A third researcher (RCCPS) verified the eligibility of the included studies.

We assessed the heterogeneity between the 2 estimates using an interaction test. The Q test was used to assess between-study heterogeneity, and the I^2^ statistic, which expresses the percentage of the total observed variability due to study heterogeneity, was calculated [26]. The I^2^ values were set relative to zero, with values ranging from 0% to 100%, [27] where 0% indicates no heterogeneity and 25%, 50%, and 75% indicate low, moderate, and high heterogeneities, respectively [26,27,28].

It is noteworthy that the study protocol for this systematic review has been published elsewhere [29] in order to ensure transparency and methodological rigor, as recommended by the Cochrane Collaboration.

### 2.3. Role of the Funding Source

There was no funding source for this study. The first author and the corresponding author had full access to all the study data.

## 3. Results

The search strategy yielded 1184 studies: 960 from the databases, 35 from clinical trial records, and 189 from additional sources. After the exclusion of 81 duplicates using EndNote™, 914 studies were selected for the title and abstract selection process. Most studies were excluded (804) based on pre-established inclusion and exclusion criteria. Among the 110 studies retained, 87 were excluded because they did not address the guiding question of the review, resulting in 23 articles for full, exhaustive reading. Among the twenty-three eligible studies, seven were excluded because they did not address the three primary endpoints simultaneously (safety, efficacy, and immunogenicity). Thus, 16 studies (5 RCTs and 11 NRCTs) were selected for data extraction, methodological evaluation, and quantitative analysis (Figure 1). None of the 189 studies from the additional sources were included in this review due to them not answering the research question.

### 3.1. Characteristics of the Studies

Table 2 chronologically summarizes the characteristics of the studies included in this systematic review. The studies were published between 2003 and 2020 [30,31,32,33,34,35,36,37,38,39,40,41,42,43,44,45]. Most studies were conducted in the USA [30,31,34,35,41,42,43,44,45] and were NRCTs (phase I) [30,35,37,39,40,41,43,44]. None of the included studies were phase III trials. The total number of study participants among the included studies was 672 patients, and the samples ranged from 7 to 167 patients. The age of the patients included in the studies ranged from 19 to 50 years. Most studies did not use a control group [30,34,37,39,40,41,43,44,45]. In addition, a predominance of placebo administration was observed in the trials that used a control group [31,33,38,42]. Regarding the follow-up time, most studies ranged from 9 to 36 weeks [30,31,32,33,35,38,40,41,42,43,44,45], and four studies had a follow-up period of 1 year or longer [34,36,37,39].

The data related to the main characteristics of the vaccines and their respective protocols used in each of the included studies are provided in Appendix A**.** The most used vaccine type was a DNA vaccine (n = 7; 43.7%) [30,31,35,40,42,43,45], followed by peptide vaccines (n = 4; 25%) [37,38,41,44], recombinant viral vectors (n = 3; 18.7%) [32,33,36], and recombinant bacterial vectors (n = 2; 12.5%) [34,39]. The most commonly used route of administration among the studies was intramuscular (n = 7; 43.7%) [30,31,35,40,42,43,45]. The most commonly used antigen for vaccine design was HPV 16 E7 oncoprotein (n = 13; 81.2%) [30,31,33,34,35,36,37,39,40,41,42,43,45].

### 3.2. Risk of Bias

The internal validity and risk of bias of the RCTs were assessed using the revised Cochrane RoB 2 [24] (Figure 2). Among the five RCT, two (40%) had a low risk of bias [31,42], the others two [33,38] had a high risk of bias, and one had some concerns [45]. Only two RCTs [31,42] appropriately described the method to generate the randomization sequence, allocation confidentiality, and the blinding of the participants and the team involved.

The analysis of the risk of bias in non-randomized studies of intervention using the ROBINS-I tool [25] (Table 3) indicated that 10 of the 11 studies had a serious risk of bias and that only 1 study [43] had a moderate risk. The studies were classified as such mainly due to participant selection bias. In addition, all eleven trials had a moderate risk of bias because they were not randomized, and five trials had a risk of bias because they did not adequately report the dropouts and/or missing data [34,40,41,43,44], in addition to not properly describing allocation confidentiality or the blinding of the participants and the team involved. Of these studies, only two [35,39] reported blinding.

### 3.3. Endpoints

Overall, it can be inferred that most patients who had adverse events related to the use of vaccines experienced mild or moderate events. There were no serious, i.e., grade 3 or higher, adverse events. In addition, there were no deaths associated with vaccine administration. The adverse events improved spontaneously, and in most studies, there were no losses associated with adverse events, except in two trials [38,42], one of which [38] was stopped prematurely. The symptoms were more associated with local events, such as pruritus, oedema, erythema, and pain than with the injection and included some systemic signs, such as flu symptoms, headaches, fatigue, and nausea. Most authors concluded that the vaccines were safe and well tolerated.

Most studies included patients with CIN 2/3 to evaluate the efficacy of therapeutic vaccines, except for three studies [39,40,45] that included only patients with CIN 3. All the included studies evaluated, at different time points, the histological regression of patients who received the vaccine. However, seven studies [34,35,38,39,40,43,44] did not evaluate lesion size regression. Viral clearance was evaluated by most studies, except for two trials [35,39]. Of the sixteen studies selected, four [31,42,44,45] correlated the efficacy of vaccines with the age of the patients, and only two trials [30,31] correlated the results with smoking and the use of oral contraceptives.

In general, the trials evaluated T cell response, antibody generation, cytokine and/or chemokine synthesis, and late hypersensitivity. Most T lymphocyte responses against HPV (n = 14) were obtained from peripheral blood mononuclear cells and/or plasma [30,31,33,34,35,37,38,39,40,41,42,43,44,45]; five of these studies analyzed tissue or cervical secretion samples [38,39,41,42,43], and one of the trials exclusively analyzed the cells infected by HPV [32]. The generation of antibodies against HPV was analyzed in seven of the sixteen studies [30,32,33,35,36,40,42]. For the analysis of T cell subsets, the authors assessed CD4+ T cells, CD8+ T cells and Th1, Th2, and Treg cells more frequently [33,34,40,41,42,43,44,45]. Only two studies performed late hypersensitivity tests [33,38]. Cytokine and/or chemokine synthesis was evaluated in four of the fifteen selected trials [38,40,42,44]. Most immunogenicity trials (n = 15; 93.7%) assessed HPV 16 [30,31,32,33,34,35,36,37,38,39,41,42,43,44,45] and HPV 18 [32,42,45], specifically the oncoproteins E6 [33,35,36,38,40,42,43,44,45] and/or E7 [30,31,33,34,35,36,37,38,39,40,41,42,43,45]. Only three studies (20%) evaluated oncoprotein E2 [32,33,40], and one trial evaluated L1 and E [33].

Most of the included clinical trials (n = 12; 75%) reported favorable and significant results for HPV-specific T cell responses after vaccination [30,31,32,33,37,38,39,40,41,42,44,45]. Among the seven studies that evaluated the production of antibodies against HPV, three showed positive results regarding the generation of a humoral response [32,33,42], and four other studies showed nonsignificant results [30,35,36,40]. The positive correlation between immunogenicity and clinical response was significant in seven of the twelve studies that demonstrated relevance in the immunological analysis [32,37,39,40,42,44].

Figure 3 briefly illustrates the main endpoints of therapeutic vaccines evaluated in this systematic review.

Since the therapeutic vaccines were heterogeneous regarding their formulation, dose, intervention protocol, and routes of administration, making meta-analysis unfeasible, therefore, we considered it more appropriate to present a qualitative synthesis of the data.

## 4. Discussion

To the best of our knowledge, this is the first systematic review to critically evaluate evidence from clinical trials simultaneously taking into account the three endpoints (safety, efficacy, and immunogenicity) of therapeutic vaccines for the treatment of patients with high-grade CIN associated with HPV. In addition, the trials investigating therapeutic vaccines reported promising results for the treatment of these lesions; however, a greater understanding of the kinetics of the immune response and of how to prevent viral escape mechanisms is required.

Overall, the trials have shown good safety and tolerability with respect to these vaccines [30,31,32,33,34,35,36,37,38,39,40,41,42,43,44,45]. Most patients had adverse reactions, but they were classified as mild or moderate; there were no grade 3 or more adverse reactions. There were no cases of death, and most participants did not discontinue the treatment due to adverse events. The adverse events most associated with vaccines were pruritus at the injection site and some systemic symptoms similar to flu symptoms.

Recent studies have also demonstrated the safety and tolerability of prophylactic HPV vaccines [46,47,48,49]. Such vaccines generate neutralizing antibodies against the proteins that form virus capsids, preventing pathologies that may originate from these viruses. This parameter is also frequently evaluated for therapeutic vaccines. Other researchers [50] determined the toxicity, safety, immunogenicity, and efficacy of the subcutaneous HPV16 SLP vaccine in 20 patients with advanced or recurrent HPV16-induced gynecological carcinoma. The authors concluded that the vaccine was well tolerated, as determined by the absence of systemic toxicity above grade II and the presence of only transient flu-like symptoms. They associated these data with favorable immunogenicity by generating a broad T cell response associated with IFNγ, TNFα, IL-5, and/or IL-10 in 84.6% of the patients. However, the clinical response did not induce tumor regression or prevent a progressive disease, demonstrating the need to administer these vaccines combined with conventional treatments such as chemotherapy. Bagarazzi et al. [14] conducted a phase I trial with 18 women previously treated for CIN 2/3 and described promising results related to the safety, tolerability, and immunogenicity of the VGX-3100 vaccine synthesized with HPV 16/18 and administered via in vivo electroporation. In addition, the immunization was well tolerated, with reports of mild reactions at the injection site and without severe or grade 3 and 4 adverse events, and no dose-limiting toxicity was observed. In addition to providing safety data, the study performed flow cytometry analysis, where it was possible to identify the induction of HPV-specific CD8+ T cells associated with granzyme B and perforin, which exhibited complete cytolytic functionality at all the doses tested. These data indicate that this vaccine was able to generate robust immune responses to high-risk HPV antigens, favouring the elimination of infected cells and the subsequent regression of the dysplastic process [14].

Efficacy was assessed via lesion regression, histopathological regression, and/or viral clearance in most studies [30,31,32,33,34,35,36,37,39,40,41,42,43,44,45]. Thus, favorable immunogenicity was observed in most of the included studies [30,31,32,33,37,38,39,40,41,42,45], even if preliminary data were presented. These data are consistent with the findings of a multicenter, double-blind phase II RCT in which safety and efficacy were evaluated in 192 women with CIN 2/3 (129 who received the vaccine, and 63 in the control group); there were significantly higher histological resolution and viral clearance rates in the vaccinated group than there were in the control group [51]. The data related to the complete histological resolution of CIN 2/3 in the sixth month showed that 18% of women monoinfected with HPV 16 who received the vaccine exhibited resolution (95% CI: 8–28%); in the placebo group, only 4% (95% CI: 0–11%) exhibited the same outcome, indicating an 80% vaccine efficacy (95% CI: 67–88%). The same outcome evaluated in a subgroup of women infected with HPV 16 and any other type of high-risk HPV showed the complete resolution of CIN 2/3 in 18% (95% CI: 4–32%) of women in the vaccine group and 8% (95% CI %: 7–24%) of women in the placebo group, representing an efficacy of 53% (95% CI: 47–61%). The complete resolution of CIN 2/3 occurred in 35% (95% CI: 21–49%) of the vaccinated patients infected with any high-risk HPV type, except HPV 16, and in 17% (95% CI: 2–32%) of patients in the placebo group, a vaccine efficacy of 52% (95% CI: 38–66%). The clearance of viral DNA from all the patients with CIN 2/3, regardless of the type of HPV, was significantly higher in the groups that received the vaccine than it was in the control group (*p* = 0.01). In addition, likewise our systematic review, the authors reported that the vaccine was well tolerated, with the most common adverse events being reactions at the injection site [51].

The safety, efficacy, and immunogenicity of a therapeutic vaccine (ISA101, associated or not with imiquimod) were also evaluated by Van Poelgeest et al. [52] in a multicenter RCT with 43 patients with vulvar and vaginal intraepithelial neoplasia. In the trial, clinical responses induced by the vaccine, in a period of 3 months, were observed in 18 of 34 patients (53%) (95% CI: 35.1–70.2), and at 12 months after vaccination, the same parameter was observed in 15 of 29 patients (52%) (95% CI: 32.5–70.6), of whom 8 had a complete histological response. Clearance occurred in all the patients. An immune response mediated by CD8+ T cells was observed in all the patients and was significantly stronger in the patients with complete responses, indicating a correlation between the immune response with the clinical response and efficacy for the treatment of high-grade vaginal and vulvar neoplasms associated with HPV 16. However, although the efficacy and immunogenicity data are promising, the safety of the vaccine needs to be better evaluated because 18% of patients in each group had symptoms, for example, allergic reactions most likely associated with the peptide used in the study, and long-term reactions at the injection site (with oedema still present after 12 months). Ten patients experienced severe adverse events (such as the development of ulcers at the injection site, which in some cases, required special interventions). To reduce adverse events in future studies, the authors suggested the use of alternative adjuvants to replace the one used in the study (Montanide), dose–response studies, and vaccination combined with imiquimod on the lesion.

Another study that showed promising results related to the efficacy and immunogenicity of therapeutic vaccines was conducted using a peptide vaccine combined with Freund’s incomplete adjuvant in 20 women with high-grade HPV-16-positive vulvar intraepithelial neoplasia [53]. The results of the study [53] indicated local adverse events, such as oedema, in 100% of the patients and systemic events, such as fever, in 64% of the patients; however, none of the adverse events exceeded grade 2, and these symptoms improved within 3 months after the last vaccination. In this same period, twelve of twenty patients (60%) (95% CI: 36–81) showed clinical responses; of these, five women showed complete lesion regression, and four showed complete HPV-16 clearance. In the follow-up period, i.e., 12 months after vaccination, 15 of the 19 patients (79%) had clinical responses (95% CI: 54–94), and 9 (47%) had a complete response (95% CI: 24–71) that was maintained at 24 months of follow-up. All the patients developed vaccine-induced T cell responses. The post hoc analyses suggested that the five patients who presented a complete response at 3 months had a significantly stronger CD4+ response and a broader response to interferon-γ CD8+ T cells than those in the patients without a complete response, demonstrating that there may be a correlation between the clinical response and immunogenicity [53].

One of the most promising studies [54] regarding the development of therapeutic vaccines against HPV was a phase III RCT in which the authors evaluated the safety, efficacy, and immunogenicity of the MVA E2 recombinant vaccine to treat intraepithelial lesions associated with HPV infection. For this, the trial evaluated 1176 women and 180 men who received the vaccine directly in their uterus, urethra, vulva, or anus. The results showed that 89.3% of female patients had complete lesion elimination after treatment with MVA E2 and that another 2.4% with CIN I exhibited histopathological regression. In the men, all the lesions were eliminated. Efficacy could also be assessed according to total HPV DNA clearance after treatment in 83% of all the patients included in the study. In addition, the vaccine did not present significant adverse events, and an excellent immune response was observed through the development of antibodies against HPV and the generation of a specific cytotoxic response against cells transformed by HPV. These data suggest that therapeutic vaccination with the MVA E2 vaccine is promising to promote immunogenicity and efficacy, in addition to demonstrating safety for the synthesis of therapeutic vaccines against HPV when applied locally [54].

In a phase II RCT, 19 patients with HPV-16-associated vulvar intraepithelial neoplasia (grade 2/3) were administered imiquimod followed by TA-CIN vaccination [55]. The results of that trial showed that complete histological regression occurred in 32% of the patients evaluated on week 10, increasing to 58% on week 20 and to 63% on week 52. The clearance of HPV 16 was evaluated on week 52, when it was possible to determine that 36% of the lesions had been cleared. On week 20, there was a significant increase in the local infiltration of CD8 and CD4 T cells in the responder patients who showed lesion regression. In contrast, the non-responder patients who had histological lesions showed an increase in regulatory T cells [55].

The safety, efficacy, and immunogenicity of therapeutic vaccines have also been evaluated for the treatment of cervical cancer. Reuschenbach et al. [56] evaluated these parameters using a peptide vaccine in 26 patients with advanced cancers over a period of 6 months. The study did not show severe adverse events associated with the use of the vaccine, and there were no dose-limiting toxicities. The development of CD4+ T cells was observed in 14 of the 20 patients, the presence of CD8+ T cells was detected in 5 of the 20 patients, and antibodies were detected in 14 of the 20 patients. The efficacy related to the tumor response was evaluated in 14 patients, of whom 64% had a stable disease as the best overall response, and 36% developed progressive disease. Thus, the authors [56] suggested that the vaccine induces cellular and humoral immune responses, does not cause severe toxicities, and provides promising results for the development of immunotherapy for cervical cancer.

Most of the studies reported in our systematic review evaluated immunogenicity only in peripheral blood cells and not in the tissue with the lesion. A positive association between immunogenicity and clinical efficacy was found in six studies [32,37,39,40,42,44]. However, despite showing promising results related to the safety, efficacy, and immunogenicity of therapeutic vaccines against HPV, some variables relevant to the success of cancer treatment, such as smoking, oral contraceptive use, age, and number of births, were controlled and/or correlated in a few studies, and none of them associated the results with the number of partners or age when they first had sexual intercourse [30,31,35,44]. This association may be relevant for the development of effective vaccines because all these factors can affect the progression of the disease as well as the immune response [57].

The assessment of immune cells is crucial in clinical research involving therapeutic vaccines to evaluate their efficacy by establishing the number and types of immune cells that infiltrated the tissue and blood circulation to determine the kinetics of these cells in the response against HPV, including the possible escape mechanisms of these viruses [39,58]. It is important to consider that disease progression is not solely related to viral infection, but to several genetic and environmental factors as well, as they modulate the immunological responses, inflammatory processes, and physiopathological mechanisms of many diseases, including cancer. Toll-like receptors are important markers of the innate immune response; they start the immunological response, releasing pro-inflammatory cytokines, causing the inhibition of its expression by microorganisms such as HPV that may promote events related to carcinogenesis [59,60]. Furthermore, the development of a therapeutic vaccine may be associated with several factors, such as a significant immune response, effective measures that control viral escape mechanisms, and the presence or absence of a correlation between immunogenicity and clinical efficacy and immunosuppression associated with these infections [61].

Strategies for the development of effective vaccines should be designed to overcome some limitations inherent to clinical trials, such as blocking local immunosuppression via in situ vaccination or using a therapeutic vaccine associated with antagonistic antibodies against inhibitory receptors such as CTLA-4 or agonist antibodies against costimulatory molecules such as CD137. The use of efficient adjuvants, such as electroporation, facilitates an increase in the permeability of the cell membrane and the consequent release of antigens and causes damage at the injection site by acting as an adjuvant and promoting the inflammatory response, the depletion of Tregs through the use of anti-CD25 antibody, and the use of TLR agonists and simultaneous administration via systemic and intralesional routes [62,63]. The intralesional route induces a more intense recruitment of intraepithelial CD8+ T cells than other routes of administration do [43], and this association with other routes of administration, such as the intramuscular route, is beneficial for the immunogenicity of therapeutic vaccines.

Since CIN 2/3 are precursor lesions of cervical cancer that have high rates of recurrence, morbidity, and mortality, the best immunological and clinical responses are observed in patients with precursor lesions. This fact is linked to systemic and local changes associated with cancer and might be related to deleterious effects on immunocompetent T cells [64,65]. Besides the strategies to produce therapeutic vaccines included in the Results section, it is noteworthy the promising results from pre-clinical trials such as those of oncolytic virus which promote tumoral reduction and significantly improve the immunological system. Such studies might also be considered as venues for effective results against cervical cancer [66,67,68].

Taken together, the studies conducted with therapeutic vaccines report promising results for the treatment of these lesions; however, a greater understanding of the kinetics of the immune response and how to prevent viral escape mechanisms and improve the associated immunogenicity and clinical response in a safe manner is required. Our study confirms the need for future RCTs, especially phase III RCTs, with a high methodological quality (i.e., larger samples, randomized, double-blind, placebo-controlled phase III, and longer follow-ups with less attrition) and standardization to enable precise comparison. We recognize some limitations of our study. First, the methodological limitations of the articles included in the analysis may have affected the outcomes, and therefore, the data should be interpreted with caution. Second, most of the studies were phase I and II trials. Third, there were significant differences in the protocols, doses, types of vaccine, definitions of clinical response, virologic clearance, and follow-up times among the studies, making meta-analysis unfeasible. Another limitation of the present systematic review is due to the use of very rigorous selection criteria, i.e., including the three endpoints (safety, efficacy and immunogenicity) simultaneously, which made us exclude large and important trials in this field from the sample, which addressed one or two of the outcomes (although such trials were addressed in the Discussion section due to their great contribution to this area of knowledge). Finally, we recommend that future systematic reviews in this field take into account at least two of the outcomes reported here in order to expand the sample of potentially included studies for evaluation.

## 5. Conclusions

In summary, we conclude that vaccines under development for the treatment of high-grade CIN (2/3) are safe and well tolerated and that most triggered only mild or moderate systemic symptoms, which spontaneously improved in a short time. The authors reported promising results for the variables related to efficacy and immunogenicity, both in the activation of T cells and development of HPV-specific antibodies and in significant results related to lesion regression, viral clearance, and/or histopathological lesion regression. These findings should be interpreted with caution because they cannot yet be considered conclusive, as most studies had a small sample size, had a low methodological quality (most of the studies were NRC and/or phase I or II RCTs), and had a relatively short follow-up period.

Thus, there is a need for future well-designed large-scale phase III RCTs with a high methodological quality, with a follow-up period of more than 1 year and with strategies to control confounding variables that may interfere with the endpoints. In addition, future studies should focus on approaches that have been underexplored, but that have yielded favorable results with respect to the other neoplasms associated with HPV. Such approaches include, for example, other routes of administration, such as administration at the lesion site, and the inclusion of other high-grade HPV types to better understand viral escape mechanisms and the development of immune responses associated with clinical efficacy.

## Figures and Tables

**Figure 1 cancers-16-00672-f001:**
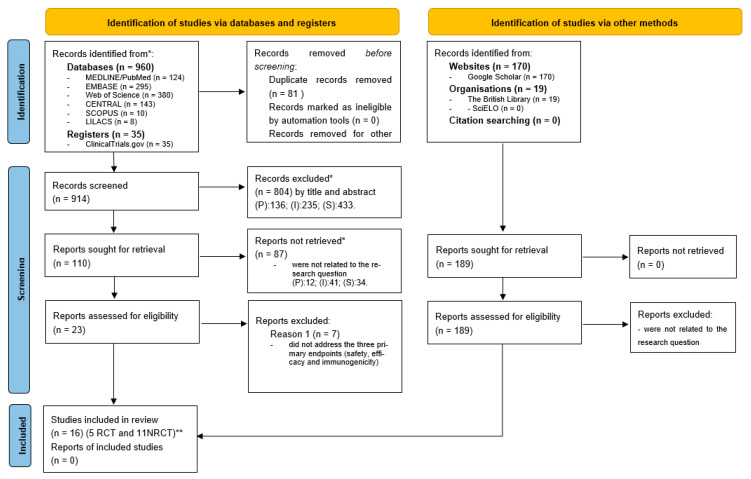
PRISMA flowchart for study selection. * Reasons for exclusion; (P) population: NIC I patients, patients with immunosuppression associated with HPV, pathologies associated with HPV other than cervical intraepithelial, studies with male subjects, and studies with animals; (I) intervention: vaccination program or prophylactic vaccine; (S) study design: reviews, specialist opinions, theses, dissertations, and observational studies. ** RCT = Randomized Controlled Trial; NCRT = Non-Randomized Controlled Trial.

**Figure 2 cancers-16-00672-f002:**
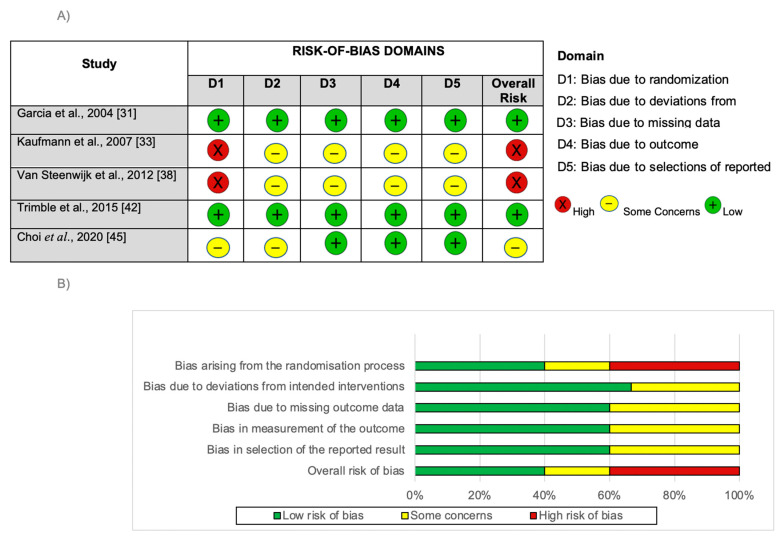
Summary of risk-of-bias judgements of Randomized Controlled Trials included according to the revised Cochrane Risk-of-Bias tool for randomized trials (RoB 2). (**A**) Internal validity and risk-of-bias assessment of clinical trials according to the RoB 2. (**B**) Percentage of risk of bias among clinical trials according to the domains of the revised Cochrane Risk-of-Bias tool for randomized trials (RoB 2). Plus symbol (+) indicates low risk of bias; negative symbol (−) indicates some concerns; (X) indicates high risk of bias. Two reviewers gave identical assessments in each domain in an independent manner.

**Figure 3 cancers-16-00672-f003:**
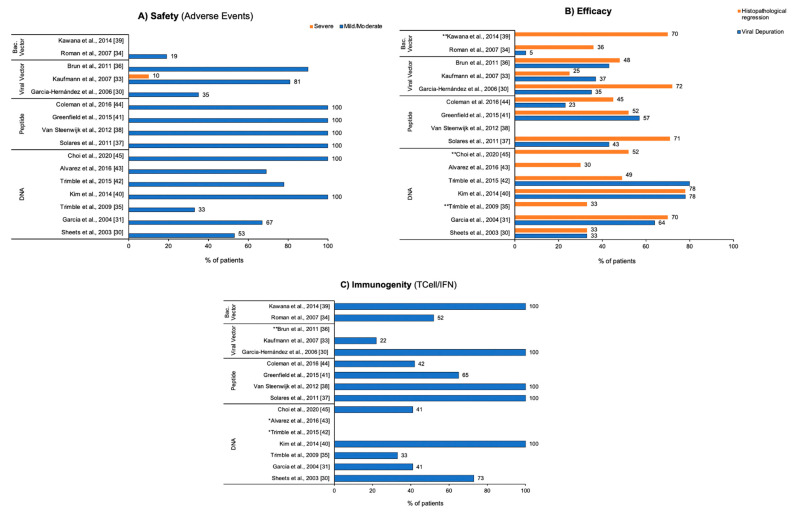
Efficacy, immunogenicity, and safety of the therapeutic vaccine used in the studies. The most frequent parameters in the studies are expressed in % of patients who presented them. (**A**) Panel Safety compares mild/moderate and severe adverse events, considering the % of patients presenting the most frequent events of the category. (**B**) Panel Efficacy compares viral reduction and histological regression. (**C**) Panel Immunogenicity compares Tcell/IFN responses. * Data presented in the study did not allow precise percentage descriptions. ** Parameter not evaluated in the study.

**Table 1 cancers-16-00672-t001:** PICOS strategy.

Acronym PICOS	Inclusion Criteria	Exclusion Criteria
P—Population	Patients with high-grade CIN (grade 2/3 associated with HPV)	Patients with associated immunosuppression
I—Intervention	Patients receiving therapeutic vaccines for the treatment of high-grade CIN 2/3 associated with HPV	Use of prophylactic vaccines for the treatment of CIN 2/3 associated with HPV or other neoplasms
C—Control	Patients who received placebo or patients serving as their own control	
O—Outcomes	Safety, efficacy, and immunogenicity of therapeutic vaccines used in patients with high-grade CIN 2/3 associated with HPV	Studies that do not report at the same time the three endpoints (safety, efficacy, and immunogenicity)
S—Study Design	RCT or NRCT	Reviews, theses, dissertations, expert opinions, editorials, protocols, clinical guidelines, and conference proceedings

**Table 2 cancers-16-00672-t002:** Characteristic and primary endpoints (efficacy, safety, and immunogenicity) of the studies included in the systematic review.

Citation/Design Country	Study Characteristics						Conclusion
	Sample	Vaccine Type/Immunogen	Adverse Events (AEs)	Virological Response	Histopathological Regression/Lesion Size	Immune Response	
Sheets et al. 2003 [30] NRCT USA	N = 15 Age: 19–44 Groups: Vaccine: 15 Placebo: 0	− DNA vaccine (ZYC101) − HPV 16 E7	AEs present in 53% of patients, the most common reaction were erythema, discomfort or other mild or moderate reactions.	Samples from responder patients were negative for HPV after vaccine treatment.	33% of patients had complete histological responses and complete response to ZYC101 in lesion size.	73% of patients had significant HPV-specific T cell response after vaccination and 87%, when considering the follow-up period.	The vaccine is associated with complete histological response with decrease in lesion size in 33% of patients, immune response in 73%, and no serious adverse events.
Garcia et al. 2004 [31] RCT USA	N = 161 Age: not reported Groups: Vaccine: 111 Placebo: 50	− DNA vaccine (ZYC101a)/HPV 16 & 18 − E6 & E7	The most common adverse events were related to the injection site, classified as mild to moderate with no major systemic side effects reported	ZYC101a induced clearance to HPV in subjects with HPV-16 or HPV-18 as well as subjects with other HPV types had higher clearance rates than matched patients who received placebo (64% versus 22% and 73% versus 25%, respectively)	The resolution of CIN 2/3 in the subgroup of women younger than 25 years was a statistically higher disease resolution rate for subjects treated with ZYC101a compared with placebo (70% versus 23%, respectively). The proportion of subjects within each treatment group without colposcopically visible lesions increased slowly but consistently, from 0% at baseline to 35–40% at the time of LEEP. The patients <25 years tended to have smaller lesions	Increased HPV-specific T cell response in patients <25 years was found in 12 patients (37%), and in patients ≥25 years, this percentage was 45%.	The vaccine was shown to be safe and well tolerated in all patients. The data found in the study support the continued clinical development of ZYC101a for the treatment of CIN 2/3 in women <25 y.o.
Garcia-Hernández et al., 2006 [32] NRCT Mexico	N = 54 Age: average 35 years old Groups: Vaccine: 34 Conization: 20	− Recombinant viral vector vaccine MVA- E2	Only a few moderate events were observed, the most frequent being headache, flu symptoms, fever, chills, moderate abdominal pain, and joint pain.	DNA viral load was significantly reduced in patients treated with MVA E2. Twelve of thirty-four patients efficiently eliminated all the HPV DNA. In 5 patients, the viral load decreased by 95%. In the other patients, the viral load was reduced between 15 and 50%. None of the 20 patients in the control group treated by conization eliminated HPV. Conization cleared the lesions in 80%, but the patients did not clear HPV.	Three weeks after the end of treatment, 56.25% of the patients with CIN 3 and two with CIN 2 were free of lesions. In 11 patients, the lesion was reduced to 50% of its original size. In 2 other patients, the CIN 3 lesion was reduced to CIN 2 and in another, the CIN 3 lesion was reduced to CIN 1. In addition, through colposcopy, 55.8% of the patients did not show the presence of HPV infection and the lesion was diagnosed as having been reduced by 100%. In 32.44% of patients, the lesion was reduced by up to 60%.	All patients developed antibodies against the MVA E2 vaccine and developed a specific cytotoxic response against papilloma-transformed cells. All patients treated with MVA E2 developed cytotoxic T lymphocytes (CTL) directed against tumor cells and the presence of CLT was correlated with lesion clearance	The vaccine can be considered safe and is a very promising candidate for the treatment of cervical lesions induced by high-grade CIN 3 HPV. The treatment leads to the elimination of the lesion as well as the elimination of viral DNA, leaving patients with better protection against future recurrences due to HPV reinfection.
Kaufmann et al. 2007 [33] RCT Germany	N = 39 Age: 20–38 Groups: Vaccine: 26 Placebo: 13	− Recombinant viral vector vaccine − HPV 16 L1E7 CVLP	Patients reported mild-to-moderate adverse events at the injection site, such as pain, induration, and itching. Reported systemic reactions were symptoms of flu and fatigue. Most of all AEs were fully recovered by the end of the study. The second AE most associated with treatment was headache.	After 48 weeks of treatment, six patients (37%) in the HPV 16 L1E7 vaccine group were HPV16 DNA negative, whereas only 1 of the placebo patients (14%) became HPV 16 DNA negative.	Histological regression to CIN 1 or normal was observed in 39% (9/23) of patients who received the vaccine and in only 25% (3/12) of patients in the placebo group. No statistically significant differences were found between the treatment and control groups with a reduction in lesion size greater than or equal to 50%	None of the patients in the placebo group had increased antibody titers, in the vaccine group a significant increase in L1-specific antibodies was observed. Measurement of isotypes showed induction of IgG (all patients), IgM (low dose 7/12; high dose 12/12) and IgA (low dose 11/12; high dose 10/12). T cell response after vaccination against E7 antigen (5 of 23 patients) was observed.	The vaccine had a very good safety profile, with only minor adverse events attributable to immunization, suggesting that it is safe and well tolerated. Antibodies with high titers against HPV 16 L1 and cellular immune responses were observed, and a trend of clinical efficacy highlighting the potentially therapeutic characteristic of this tested strategy
Roman et al. 2007 [34] NRCT USA	N = 21 Age: average 26 years old Groups: Vaccine: 21 Placebo: 0	− Recombinant bacterial vector vaccine (SGN-00101)/ − HPV 16 E7	No grade III or IV toxicities were observed. There were four women who had Grade 2 injection site reactions which were of short duration (lasting less than a week).	Viral clearance occurred in only 1 woman. HPV clearance was not associated with lesion regression or immune response.	Seven of the twenty women (35%) evaluated for response had complete regression of their intraepithelial neoplasia at the time of LLETZ, one (5%) regressed to CIN I, eleven (55%) had a stable disease, and one (5%) progressed due to a worsening injury. Of the 17 women who completed 1 year follow-up after LLETZ, 13 (77%) remained without evidence of recurrent CIN at their last follow-up, and 4 of 13 women (31%) were PCR negative for HPV at the end of the study.	52% of patients had evidence of an immune response to at least one peptide, suggesting that the vaccine was immunogenic in women with high-grade CIN and HPV infection.	Vaccine was considered safe and well tolerated. The HPV clearance appeared to be limited and generated modest levels of immunity and clinical response in patients with high-grade CIN. Although, the small number of patients evaluated, and the known spontaneous regression rate of CIN preclude any definitive conclusions as to the usefulness of the vaccine that has been tested
Trimble et al. 2009 [35] NRCT USA	N = 15 Age: 18–50 Groups: Vaccine: 15 Placebo: 0	− DNA vaccine (pNGVl4A-Sig/E7(detox)/HSP70/ − HPV 16 E7	Most adverse events were mild with transient discomfort at the injection site. Systemic symptoms after vaccination were also reported by 5 of 15 subjects.	Not reported	Complete histological regression occurred in 3/9 (33%) patients in the highest dose cohort (3 mg) on week 15. Although the difference is not significant, it is slightly greater than would be expected in a control cohort (25%).	Vaccination did not elicit antibody responses. Measurable titers at study entry of anti-E6 IgG antibody in 3/15 (20%) and anti-E7 IgG antibody in 2/15 (13.3%) were noted. E7 titers were not increased after vaccination with E7DNA synthesis in any dose cohort.	The vaccine was safe and well tolerated.
Brun et al. 2011 [36] NRCT France	N = 10 Age: 25–44 Groups: Vaccine: 10 Placebo: 0	− Recombinant viral vector vaccine TG4001 − HPV 16 E6 & E7	90% of patients reported some local and systemic adverse event. Intensity ranged from mild to moderate, with no episodes of grade 3 local reaction	Nine of twenty-one patients showed improvement in their HPV 16-associated infection. HPV 16 mRNA clearance was associated with cytological and colposcopy regression in 7 of 10 responders. Of the 10 respondents, 8 did not have HPV 16 DNA.	48% (10 of 21) of patients responded to clinical treatment within the 6 months. They showed no or small changes in colposcopy and cytological diagnosis showed low-grade lesions or less, and 8 of them did not undergo surgery. The median times to disappearance of high-grade lesion, for HPV 16 E6 and E7 clearance, and HR-HPV DNA clearance were 13.5; 13.3, and 26 weeks, respectively.	At baseline, all patients had E7 antibody responses and 3 (19%) had E6 antibody responses. After treatment with TG4001, no patient developed or improved an antibody response to E6 or E7 as assessed by this method	The vaccine was safe and well tolerated. The results obtained in the trials of this study provided promising results for the development and further study of the TG4001 vaccine for the treatment of cervical intraepithelial neoplasia (CIN 2/3).
Solares et al. 2011 [37] NRCT Cuba	N = 7 Age: 24–43 Groups: Vaccine: 7 Placebo: 0	− Peptide vaccine CIGB-228 − HPV 16 E7	No toxicity beyond grade 3 was observed in the experiment. All patients reported local pain at the vaccination site and 6 patients reported a burning sensation.	HPV was eliminated in three of the five patients with complete response.	The colposcopic response was evidenced in 6 of the 7 patients (85.7%), 4 (57.1%) complete and 2 (28.6%) partials. Histological analyzes indicated that 57.1% of patients (4/7) had complete regression, while 14.3% (1/7) had a decreased histological grade.	Cellular immune response was observed in all patients after vaccination.	Vaccination with CIGB-228 is safe and well tolerated. Moreover, resulted in lesion regression and HPV clearance. vaccination is capable of inducing IFN-N-associated T-cell responses in women with high-grade CIN.
Van Steenwijk et al. 2012 [38] RCT Netherlands	N = 10 Age: not reported Groups: Vaccine: 5 Placebo: 5	− Peptide vaccine − HPV 16 E6/7	All 5 patients in the vaccination group experienced adverse reactions that were mainly flu-like symptoms and injection site reactions. There were dropouts associated with side effects. Study ended prematurely.	In most patients, there was no change in the viral status	In most patients there was no change in histopathological status. There was no clearance of HPV at the time of excision.	A strong IFN-associated T cell-specific response to HPV was detected in all vaccinated patients. Vaccination of patients with HSIL resulted in increased immunity to HPV 16-specific T cells. At the time of HFS treatment, HPV 16-specific IFN-γ production was found in 3/5 vaccinated patients. Three of the 4 who received placebo remained unresponsive to HPV 16 E6/E7.	The study was stopped prematurely. Suggested the development of future studies focused on the development of a better tolerated formulation. No conclusions can be drawn about vaccine-enhanced T-cell infiltration into the lesion. Overall, the study shows that the vaccine may increase the number of circulating IFN-γ-producing HPV 16-specific T cells in patients with high-grade lesions.
Kawana et al. 2014 [39] NRCT Japan	N = 10 Age: unreported Groups: Vaccine: 10 Placebo: 0	− Recombinant bacterial vector vaccine (GLBL101c) − HPV 16 E7	No patient had serious side effects induced by vaccine. No patient was withdrawn from the study due to adverse event.	Not reported	Combining the patients from Steps 1 and 2 who received four capsules/day, 7 of 10 patients (70%) had a histopathological regression to CIN2 on week 9, and 1 patient had a negative pathology grade for CIN2 on week 12. Of the 13 patients who received four–six capsules/day, 9 patients (69%) with a pathological grade lower than CIN2 did not require additional surgical treatment and were followed up cytologically. The histopathological regression for CIN2 in response to a GLBL101c regimen of four capsules/day was 80%.	Oral administration of GLBL101c predominantly induces E7-CMI from the mucosa towards the cervical epithelium	Oral administration with GLBL101c can be considered safe and well tolerated. Oral administration of E7-expressing Lactobacillus-based vaccine can induce E7-specific mucosal immunity in uterine cervical lesions. The vaccine was able to induce mucosal E7-CMI, but had no systemic response
Kim et al. 2014 [40] NRCT Korea	N = 9 Age: 23–44 groups Vaccine: 9 Placebo: 0	− DNA vaccine (pGX-188E) − HPV 16 and 18, E6 and E7	AEs were considered mild (grade 1) and all patients fully recovered within three days of vaccination.	Until week 36 (VF2), 7 patients eliminated the virus that had been found at the beginning of treatment (HPV 16 and/or 18) and also had lesion regression, resulting in a perfect correlation between clinical and virological responses.	Eight weeks after the last vaccination (VF1), 6 of the 9 patients were free of lesions. GX-188E vaccination led to a clinically and virologically significant complete response rate of 78%. Viral clearance (4 of 9 patients) and cytological regression (3 of 9 patients) were already apparent on week 12 and most complete responders (6 of 7) cleared cervical lesions by week 20 after vaccination	The vaccine induced a significant E6/E7-specific IFN-γ-producing T cell response in all 9 patients with CIN3. the antibody titers to E6 were not induced in any dose cohort patients after vaccination. Three of the 9 patients generated weak anti-E7 antibody responses following vaccination with antibody titers ranging from 1:8 to 1:256.	The administration of GX-188E, being considered safe and well-tolerated. The vaccination in patients with CIN 3 substantially increased both HPV-specific CD8 T cell. Although the (n = 9) is too small to reach a definitive conclusion
Greenfield et al. 2015 [41] NRCT USA	N = 24 Age: 22–42 Groups: Vaccine: 24 Placebo: 0	− Peptide vaccine (Pepcan) − HPV 16 E6	The most common AEs reported were immediate responses related to the injection site with no signs of toxicity.	At least one HPV type present at entry became undetectable in 13 of 23 (57%) patients. Per dose, rates were 83%, 50%, 50% and 40%, with the highest undetectability at the lowest dose	The best histological response was seen at the 50 µg dose with a regression rate of 83% (n = 6), and the overall rate was 52% (n = 23). CIN 2/3 was no longer present in 9 of 23 (39%) patients who completed the study (complete responders), and CIN 2/3 lesions measured ≤0.2 mm^2^ in 3 (13%) patients (partial responders). Five of the 12 patients with no visible lesions after vaccination were histological nonresponses with persistent CIN 2/3.	Th1 cells were significantly increased after four vaccinations. New CD3 T cell responses and at least one region of the E6 protein were detected in 15 of 23 patients (65%), with the increase in responses after vaccination being statistically significant in 10 patients (43%). The best CD3 T cell response rates to E6 were at doses of 50 and 250 µg (83%).	The PepCan vaccine is safe, no signs of vaccine-related toxicity were identified. As the number of subjects in each dose group was small (n = 6), this study was not designed to show significant differences. The systemic level of Th1 cells increased significantly, suggesting that Candida, who induces interleukin-12 (IL-12) in vitro, may have an effect on Th1 promotion.
Trimble et al. 2015 [42] RCT USA	N = 167 Age: 24–41 Groups: Vaccine: 125 Placebo: 42	− DNA vaccine (VGX-3100) − HPV 16 and 18, E6 and E7	Injection site reactions occurred in most patients, however, only erythema showed a statistically significant difference between the vaccine group and the placebo group. Four patients discontinued dosing due to an adverse event. No related serious adverse events were reported	Concomitant analysis of histopathological regression with viral clearance as per protocol: 40.2% (VGX-3100 group) and 14.3% (placebo). Viral clearance was more associated with patients who received VGX-3100 (80%) than in the placebo group (50%)	Histopathological regression according to the protocol: 49.5% (VGX-3100 group) and 30.6% (placebo).	VGX-3100 induced significantly increased frequencies of activated, antigen-specific CD8+ T cells identified by cell surface expression of CD137, which also expressed perforins compared to placebo. Humoral responses were also greater in patients in the VGX-3100 group compared to those in the placebo group.	Treatment with VGX-3100 was well tolerated. The trial showed that the administration of the DNA vaccine encoded with viral proteins can trigger adaptive immune responses that have a therapeutic effect on cervical lesions. These findings suggest that VGX-3100 offers a non-surgical option for the treatment of 2/3 CIN that could change the approach to treating this very common disease
Alvarez et al. 2016 [43] NRCT USA	N = 32 Age: 20–44 Groups: Vaccine: 32 Placebo: 0	DNA vaccine (pNGVL4a-CRT/E7 (detox) − HPV 16 E7	69% of patients experienced vaccine-related adverse events. The events were more related to the injection site and were not greater than grade 1 events. No serious vaccine-related adverse events were observed.	No differences were found between pre- and post-vaccination viral loads in any of the treatment cohorts	Histological regression for CIN 1 or less occurred in 8 of 27 (30%) patients who received all vaccinations. Persistent 2/3 CIN was observed in 19 (70%) patients.	Immune responses to E7 were minimal, and not significantly different from responses to HPV 16 E6, which was not included in pNGVL4a-CRT-E7.	The vaccine was well tolerated. An increase in the specific immune response to HPV was noted. Although a local CD8+ T cell response appeared to be more robust with intralesional vaccination, none of the vaccination routes were immunogenic.
Coleman et al. 2016 [44] NRCT USA	N = 34 Age: Not reported Groups: Vaccine: 34 Placebo: 0	Peptide vaccine (Pepcan) − HPV 16 E6	No dose-limiting toxicities were observed. The most common adverse events were mild to moderate at the injection site.	Three of the 13 women in whom HPV 16 was detected early became undetectable after vaccination and was persistent in nine patients.	Histological regression rates were 50% at the 50 μg doses (7 of 14) and 100 μg (3 of 6), 33% at the 250 μg dose and 40% at the 500 μg dose, 45% in total (14 of 31).	The immunological profile revealed an increase in type 1 helper T cells after vaccinations.	The Pepcan vaccine proved to be safe and demonstrated a decrease in HPV 16 viral load as well as histological regression.
Choi et al. 2020 [45] RCT South Korea	N = 71 Age: 19–50 Groups: Vaccine: 64 Placebo: 0	DNA vaccine (GX-188E) HPV 16 AND 18—E6/E7	AE (occurring in 94.4% and 100.0% in the 1 and 4 mg GX-188E groups, respectively). None serious AEs were related to the DNA vaccine.	Not reported	Histopathologic regression occurred in 35 (67%) of the 52 patients. 77% of the patients with histologic regression showed HPV clearance.	IFN-γ ELISpot responses ≥3-fold over baseline indicated the drug was efficacious.	GX-188E was well tolerated by all the patients.

RCT: Randomized Controlled Trial; NRCT: Non-Randomized Controlled Trial.

**Table 3 cancers-16-00672-t003:** Risk-of-bias judgements of non-randomized studies of interventions via ROBINS-I.

	* Domains ROBINS-I							Overall Judgment ROBINS-I **
Study	Confounding Bias	Participant Selection Bias	Classification of Intervention Bias	Bias Due to Intervention Deviations	Incomplete Data Bias	Outcome Measurement Bias	Selective Outcome Reporting Bias	
Sheets et al. 2003 [30]	Moderate	Serious	Low	Low	Low	Low	Moderate	Serious
Garcia-Hernández et al. 2006 [32]	Moderate	Serious	Low	Serious	Low	Low	Low	Serious
Roman et al. 2007 [34]	Moderate	Serious	Low	Low	Moderate	Low	Low	Serious
Trimble et al. 2009 [35]	Moderate	Serious	Low	Low	Low	Low	Low	Serious
Brun et al. 2011 [36]	Moderate	Serious	Low	Low	Moderate	Low	Low	Serious
Solares et al. 2011 [37]	Moderate	Serious	Low	Low	Low	Low	Low	Serious
Kawana t al. 2014 [39]	Moderate	Serious	Low	Low	Low	Moderate	Moderate	Serious
Kim et al. 2014 [40]	Moderate	Serious	Low	Low	Moderate	Low	Low	Serious
Greenfield et al. 2015 [41]	Moderate	Serious	Low	Low	Moderate	Low	Moderate	Serious
Alvarez et al. 2016 [43]	Moderate	Moderate	Low	Low	Moderate	Low	Moderate	Moderate
Coleman et al. 2016 [44]	Moderate	Serious	Low	Low	Moderate	Low	Moderate	Serious

Acronyms: * ROBINS-I, Risk of Bias In Non-randomized Studies of Intervention [41]. ** The global judgement of ROBINS-I is systematized and attributed as follows: Low risk of bias, in which the study is comparable to a well-designed randomized trial (the study is considered as having a low risk of bias for all domains). Moderate risk of bias: the study is consistent with a non-randomized study design, but cannot be considered comparable to a well-designed randomized study (in this case, the study is considered as having a low or moderate risk of bias for all domains). Serious risk of bias: the study has some important problems (the study is considered as having a low or moderate risk of bias for most domains, but presents a serious risk of bias in at least one of the domains). Critical risk of bias: the study is too problematic to provide any evidence (the study is considered as having a critical risk of bias in at least one domain). No information: when no information is available to provide grounds to any judgment of the risk of bias (missing information on one or more domains) [41]. Two reviewers gave identical assessments in each domain in an independent manner.

## Data Availability

The data presented in this study are available in this article.

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
