# Peer review of "Safety, Efficacy, and Immunogenicity of Therapeutic Vaccines for Patients with High-Grade Cervical Intraepithelial Neoplasia (CIN 2/3) Associated with Human Papillomavirus: A Systematic Review"

_cancers, 2024, doi:10.3390/cancers16030672_

Round 1
Reviewer 1 Report (New Reviewer)
There is growing interest in immunotherapeutic approaches to the treatment of HPV-related lesions, either as adjuvant treatment for HPV-related cancer or as primary treatment of HPV-associated CIN 2/3 to reduce the risk of progression to cancer. The field has been active for more than 2 decades, and has tried a wide variety of HPV antigens and delivery vehicles. In this review by Goncalves, the authors summarize the safety, immunogenicity and efficacy of these vaccines. They indicate that are safe, eliciting primarily mild and transient side effects, and most stimulate a systemic immune response. However the data on efficacy are much less consistent. Interestingly of over 1,000 studies published, only 16 met all of the authors’ criteria regarding safety, immunogenicity and efficacy, and the methodologies and outcomes were too heterogeneous to allow for meta-analysis. The authors should be commended for trying to instill some rigor into the analysis by requiring safety, immunogenicity and efficacy endpoints. Unfortunately by requiring all three, they are ending up with some of the less informative studies in the field. Some of the bigger, more informative RCTs appear to be excluded on this basis, and some are described in detail in the discussion. They should be included instead in the results section.
In general, the main message of this paper, taking into account the data from the 16 studies that met all of the authors’ requirements, is that while the vaccines largely appear to be safe it is difficult to draw any conclusions about them with respect to efficacy, due to small numbers, inconsistency of outcome measures between studies, and high risk of bias. Furthermore, in the absence of interpretable efficacy data, it is impossible to interpret the meaning of the measured immune responses.
Introduction
Line 53- it is inaccurate to say that HPV is one of the “predominant causes” of CIN and cervical cancer- rather, it is necessary but insufficient.
Line 58- please use the term HPV “type” rather than HPV “subtype” here and throughout the manuscript.
Line 78- use of the word “structure” is somewhat awkward- may want to use the word “antigens” instead.
Materials and methods
Line 124- sentence beginning with “Also studies that…” is confusing.
Line 133- many of these immune correlative studies were performed with PBMC, not serum.
Line-219- conization is not really a “control”
Line-223- the list of different types of vaccines is presented as though they are mutually exclusive- this is of course not the case, and the section should be clarified by dividing it into vectors and inserts.
Line-228- oncoprotein should be singular.
Results
Table 2- It is hard to compare papers within the able because definitions of virologic, immunologic and clinical responses were highly variable.
Table 2- The table needs to be edited carefully. The first sentence of the conclusion of the first article needs to be fixed. There are other typos as well, including the word” analyzes” in the Solares article section.
Table 2- In the immunologic responses section of the last paper described (Choi et al), it says that the Elispot responses indicated that the vaccine was efficacious. Clearly some of the information in the table is not correct- was the summary of each of the sections of the table those of the authors of this manuscript or of the authors of the manuscript being cited?
Figure 2- part of section D3 is missing on the bottom.
Figure 3- the authors presumably mean viral clearance when they say “depuration”
Figure 3- It is very difficult to interpret the data in this table because definitions of virologic, immunologic and clinical responses were highly variable.
Discussion
Line-321- saying that the trials reported promising and favorable results” is probably an overstatement given the serious limitations of each these studies. Same for line 513.
Line-331- most of the information in this paragraph and the following paragraphs through line 447 belong in the results section, not the discussion. They describe some of the most important studies in the field.
Line 357- was this trial, one of the larger ones, excluded from Table 2 because it did not report on all of the required endpoints?
Line 521-It was disappointing that the authors did not call for use of standardized methodology across studies, including common definitions of clinical response, virologic clearance, follow-up periods and suitable control populations.
Overall the English language writing is good but the manuscript does require careful editing.
Author Response
- REVIEWER 1 –
Comments and Suggestions for Authors
There is growing interest in immunotherapeutic approaches to the treatment of HPV-related lesions, either as adjuvant treatment for HPV-related cancer or as primary treatment of HPV-associated CIN 2/3 to reduce the risk of progression to cancer. The field has been active for more than 2 decades, and has tried a wide variety of HPV antigens and delivery vehicles. In this review by Goncalves, the authors summarize the safety, immunogenicity and efficacy of these vaccines. They indicate that are safe, eliciting primarily mild and transient side effects, and most stimulate a systemic immune response. However the data on efficacy are much less consistent. Interestingly of over 1,000 studies published, only 16 met all of the authors’ criteria regarding safety, immunogenicity and efficacy, and the methodologies and outcomes were too heterogeneous to allow for meta-analysis. The authors should be commended for trying to instill some rigor into the analysis by requiring safety, immunogenicity and efficacy endpoints. Unfortunately by requiring all three, they are ending up with some of the less informative studies in the field. Some of the bigger, more informative RCTs appear to be excluded on this basis, and some are described in detail in the discussion. They should be included instead in the results section.
In general, the main message of this paper, taking into account the data from the 16 studies that met all of the authors’ requirements, is that while the vaccines largely appear to be safe it is difficult to draw any conclusions about them with respect to efficacy, due to small numbers, inconsistency of outcome measures between studies, and high risk of bias. Furthermore, in the absence of interpretable efficacy data, it is impossible to interpret the meaning of the measured immune responses.
Response: Thank you very much for the feedback, as well as for this timely comment and suggestions for corrections.
In fact, this is a systematic review with a clear objective and criteria previously established and protocolled, and it was carried out with great rigor, considering the three outcomes (safety, immunogenicity and efficacy endpoints) simultaneously. Since it was never addressed in a previous systematic review on the subject these three endpoints together, the present review fills this gap in the literature) which explains the small number of articles that met all the criteria. Therefore, we were careful to discuss the trials that were excluded from the eligibility criteria but that reported one or another important outcome.
Introduction
Line 53- it is inaccurate to say that HPV is one of the “predominant causes” of CIN and cervical cancer- rather, it is necessary but insufficient.
Response: Correction performed. Thank you!
Line 58- please use the term HPV “type” rather than HPV “subtype” here and throughout the manuscript.
Response: Correction done. Thanks!
Line 78- use of the word “structure” is somewhat awkward- may want to use the word “antigens” instead.
Response: Absolutely. Thanks for suggestion.
Materials and methods
Line 124- sentence beginning with “Also studies that…” is confusing. .
Response: Correction done. Thanks!
Line 133- many of these immune correlative studies were performed with PBMC, not serum.
Response: Correction performed. Thank you!
Line-219- conization is not really a “control” –
Response: We agree with you. However, it was considered a “control” by the authors of the trial. Thus, we removed this sentence a per suggested.
Line-223- the list of different types of vaccines is presented as though they are mutually exclusive- this is of course not the case, and the section should be clarified by dividing it into vectors and inserts.
Response: Thank you for this comment. Indeed, we agree with you, but we just know exactly how such vaccines were reported and classified in the trials by the authors
Line-228- oncoprotein should be singular.
Response: OK done. Thanks!
Results
Table 2- It is hard to compare papers within the able because definitions of virologic, immunologic and clinical responses were highly variable.
Response: You are absolutely correct and we are in line with you. However, this is one of limitations of the present systematic review. We have added this limitation in the last paragraph of the discussion.
Table 2- The table needs to be edited carefully. The first sentence of the conclusion of the first article needs to be fixed. There are other typos as well, including the word” analyzes” in the Solares article section.
Response: Correction done. Thank you for your careful review.
Table 2- In the immunologic responses section of the last paper described (Choi et al), it says that the Elispot responses indicated that the vaccine was efficacious. Clearly some of the information in the table is not correct- was the summary of each of the sections of the table those of the authors of this manuscript or of the authors of the manuscript being cited?
Response: We have performed the corrections according to the data presented in the trial.
Figure 2- part of section D3 is missing on the bottom.
Response: Figure corrected.
Figure 3- the authors presumably mean viral clearance when they say “depuration”
Response: Figure corrected
Figure 3- It is very difficult to interpret the data in this table because definitions of virologic, immunologic and clinical responses were highly variable.
Response: We agreed with you. However, this is one of limitations of the present systematic review. Thus, we have pointed out this limitation in last paragraph of the discussion.
Discussion
Line-321- saying that the trials reported promising and favorable results” is probably an overstatement given the serious limitations of each these studies. Same for line 513.
Response: OK. We've made adjustments to the wording and removed the word "favorable", keeping only "promising results".
Line-331- most of the information in this paragraph and the following paragraphs through line 447 belong in the results section, not the discussion. They describe some of the most important studies in the field.
Line 357- was this trial, one of the larger ones, excluded from Table 2 because it did not report on all of the required endpoints?
Response: Exactly for this reason. This trial did not report the immunogenicity endpoint along with the other two endpoints (safety and efficacy). We would like to point out that our review had this criterion established from the beginning. Therefore, it is not possible to transpose as well as include it in the results section, as we would be violating the rigor of the method. In order to balance this issue raised, we have added this issue as a limitation of the review.
“Another limitation of the present systematic review is due to the fact that we used very rigorous selection criteria, i.e., included the three endpoints (safety, efficacy and immunogenicity) simultaneously, which made us exclude large and important trials in this field from the sample, which addressed one or two of the outcomes (although such trials were addressed in the discussion section, due to their great contribution to this area of knowledge).”
In addition, we recommend that future systematic reviews in this field take into account at least two of the outcomes reported here in order to expand the sample of potentially included studies for evaluation.
Line 521-It was disappointing that the authors did not call for use of standardized methodology across studies, including common definitions of clinical response, virologic clearance, follow-up periods and suitable control populations. .
Response: Changes performed.
Comments on the Quality of English Language
Overall the English language writing is good but the manuscript does require careful editing.
Response: English language was revised.
Reviewer 2 Report (Previous Reviewer 1)
This was a very well-done systematic review of the published studies utilizing therapeutic vaccines for patients with CIN2/3.
The methodology was sound and well-described.
The authors described the limitations of each study and the conclusions based on these studies.
Minor edits.
Line 313 rewording of sentence needed
Line 427 missing the V in HPV
Very well written, only minor suggestions- listed above
Author Response
- REVIEWER 2 – Minor revision
Comments and Suggestions for Authors
This was a very well-done systematic review of the published studies utilizing therapeutic vaccines for patients with CIN2/3.
The methodology was sound and well-described.
The authors described the limitations of each study and the conclusions based on these studies.
Response: Thank you so much for your positive feedback regarding our paper.
Minor edits.
Line 313 rewording of sentence needed
Response: Correction done. Thanks!
Line 427 missing the V in HPV
Response: Correction done. Thanks!
Comments on the Quality of English Language
Very well written, only minor suggestions- listed above
Reviewer 3 Report (New Reviewer)
To address whether the therapeutic vaccination could be an effective measure to treat high-grade cervical intraepithelial neoplasia (CIN 2/3), the authors analyzed reported clinical trials over the diverse sources. They retrieved 1,184 studies: 960 from the databases, 35 from clinical 193 trial records, and 189 from additional sources. After critically evaluating the evidence from clinical trials on the safety, efficacy, and immunogenicity of therapeutic vaccines in the treatment of patients with high-grade CIN associated with HPV, they systematically reviewed 16 trials (RCTs and NRCTs) fitting to their analysis criteria. The therapeutic vaccines analyzed in this study appeared safe with mild or moderate AEs. All the analyzed clinical trials manifested clinical efficacy with regards to lesions and histopathological regression or viral clearance. They conclude with the notion that greater investments in well-designed phase III RCT is urgently needed.
This paper comparatively summarizes multiple clinical trial data, which would provide plain information how were past clinical trials like. Overt message of this study seems to be the fact that various therapeutic HPV vaccines were not dangerous. Given the efficacy of vaccines is affected by the nature of antigens, delivery method, and adjuvants, it is a weak point of this study that any comparative analyses are lacking concerning the tested vaccines and way of immunization. In this context, this study would remain superficial in its depth of analysis.
The authors do not seem to have commands in immunology and vaccinology per se. They use awkward terms such as “late hypersensitivity” for delayed type hypersensitivity (DTH). The immunogen used for the study by Roman et al is a recombinant fusion protein consisting of M. bovis Hsp65 and E7 rather than a recombinant bacterial vector vaccine. Reading by a vaccinologist is required.
Awkward English expressions were noted throughout the text.
[Minor comments]
Abbreviations should be explained when they first come in the text.
Please check whether Table 2, being the most important information in this manuscript, contains valid information. For example, the study by Choi et al shows 67% histopathologic regression at 36 weeks, while the authors noted only 51.6% regression at 20 weeks. The results at later time points would make more importance. In Roman et al study, CD4+ responses were noted while no significant CD8+ reactivity was detected. The CD4+ responses could have had substantial effects on the outcomes.
Fig. 3 is a very excellent way of description. But having clinical stage in efficacy figure will be more helpful to compare each other. Number for safety is missing for Brun et al.
There are a number of awkward phrases in the text. Recommend English edition by a qualified editor.
Author Response
- REVIEWER 3 – Major Revision
Comments and Suggestions for Authors
To address whether the therapeutic vaccination could be an effective measure to treat high-grade cervical intraepithelial neoplasia (CIN 2/3), the authors analyzed reported clinical trials over the diverse sources. They retrieved 1,184 studies: 960 from the databases, 35 from clinical 193 trial records, and 189 from additional sources. After critically evaluating the evidence from clinical trials on the safety, efficacy, and immunogenicity of therapeutic vaccines in the treatment of patients with high-grade CIN associated with HPV, they systematically reviewed 16 trials (RCTs and NRCTs) fitting to their analysis criteria. The therapeutic vaccines analyzed in this study appeared safe with mild or moderate AEs. All the analyzed clinical trials manifested clinical efficacy with regards to lesions and histopathological regression or viral clearance. They conclude with the notion that greater investments in well-designed phase III RCT is urgently needed.
This paper comparatively summarizes multiple clinical trial data, which would provide plain information how were past clinical trials like. Overt message of this study seems to be the fact that various therapeutic HPV vaccines were not dangerous. Given the efficacy of vaccines is affected by the nature of antigens, delivery method, and adjuvants, it is a weak point of this study that any comparative analyses are lacking concerning the tested vaccines and way of immunization. In this context, this study would remain superficial in its depth of analysis.
Response: Yes indeed, the overt message of this study is that most of therapeutic HPV vaccines are safe and, consequently, were not dangerous. Regarding the call of attention to the fact that there was a lack of comparative analyzes on our part in relation to the vaccines tested and the form of immunization, it is emphasized that it was not the objective of this review.
Furthermore, due to the low number of studies included (since we are strictly following our eligibility criteria with the 3 endpoints for a more complete evaluation), from a methodological and statistical perspective, it would be impracticable to stratify results into several subgroups for comparative analyses. In fact, if we had a larger n og includee studies (which is not our case), subgroup and comparative analyzes would be quite interesting.
The authors do not seem to have commands in immunology and vaccinology per se. They use awkward terms such as “late hypersensitivity” for delayed type hypersensitivity (DTH). The immunogen used for the study by Roman et al is a recombinant fusion protein consisting of M. bovis Hsp65 and E7 rather than a recombinant bacterial vector vaccine. Reading by a vaccinologist is required.
Response: The article was fully reviewed and one of the authors has a Ph.D. in Basic and Applied Immunology.
Awkward English expressions were noted throughout the text.
Response: English language was revised.
[Minor comments]
Abbreviations should be explained when they first come in the text.
Please check whether Table 2, being the most important information in this manuscript, contains valid information. For example, the study by Choi et al shows 67% histopathologic regression at 36 weeks, while the authors noted only 51.6% regression at 20 weeks. The results at later time points would make more importance. In Roman et al study, CD4+ responses were noted while no significant CD8+ reactivity was detected. The CD4+ responses could have had substantial effects on the outcomes.
Fig. 3 is a very excellent way of description. But having clinical stage in efficacy figure will be more helpful to compare each other.
Number for safety is missing for Brun et al.
Response: We made the corrections in figure 3
Comments on the Quality of English Language
There are a number of awkward phrases in the text. Recommend English edition by a qualified editor.
Response: English language was revised.
Reviewer 4 Report (New Reviewer)
Based on the manuscript, it appears that the authors conducted a systematic review of clinical trials on the use of therapeutic vaccines for the treatment of high-grade cervical intraepithelial neoplasia (CIN) associated with human papillomavirus (HPV). The authors aimed to evaluate the safety, efficacy, and immunogenicity of therapeutic vaccines in this context.
The authors searched several databases for relevant studies and identified 16 that met their criteria. They found that the therapeutic vaccines were heterogeneous in terms of their formulation, dose, intervention protocol, and routes of administration, which made it unfeasible to perform a meta-analysis.
Overall, the text provides a clear and concise summary of the author's research and findings, highlighting the need for further research in this area. The text could be improved by providing more detailed information on the specific studies included in the systematic review, and by discussing potential limitations of the research.
- Introduction Section:
HPV is a major virus associated with the development of cervical cancer in women, as mentioned in the first and second paragraphs. Considering the significance of this virus, it is crucial to explore effective treatment options. Currently, there are various therapeutic approaches available for this case, including immunotherapy, virotherapy, chemotherapy, and more. These treatments aim to combat HPV and its associated diseases, such as high-grade cervical intraepithelial neoplasia (CIN 2/3), and offer potential solutions for patients. By investigating and evaluating these therapies, we can contribute to improving the management and outcomes of HPV-related conditions.
Therefore, I recommended some various therapeutic to use and check according to your text:
1) DOI: 10.1016/j.micpath.2020.104438
2) DOI: 10.1186/s12985-021-01571-7
3) DOI: 10.1016/j.sjbs.2021.06.043
- Material Methods section:
1) Can you provide an overview of the inclusion and exclusion criteria used in the selection of studies for this systematic review? Specifically, what specific criteria were employed to determine the eligibility of studies in terms of study design, patient population, therapeutic vaccines evaluated, and the endpoints of safety, efficacy, and immunogenicity?
2) Can you provide more information about the process of manually analyzing the references in the included studies to find additional relevant studies? Specifically, how was the manual analysis conducted, and what criteria were used to determine the relevance of the identified references?
3) Based on the given text materials and Methods, here are some important questions that can be derived:
1. What is the current understanding of the relationship between HPV and high-grade cervical intraepithelial neoplasia (CIN) as well as cervical cancer?
2. What is the current status of research on therapeutic vaccines for the treatment of high-grade CIN associated with HPV?
3. Were there any restrictions on the data or language in the studies included?
4. What were the primary endpoints assessed in the clinical trials?
5. What were the key findings regarding the safety and efficacy of therapeutic vaccines?
6. Were there any commonalities or differences observed among the therapeutic vaccines in terms of formulation, dose, intervention protocol, or routes of administration?
7. Why was a meta-analysis deemed unfeasible for the collected data?
8. Was there any correlation observed between immunogenicity and clinical response?
9. What are the implications of the findings for the development and future research of therapeutic vaccines?
10. In conclusion, what are the main recommendations put forward by the authors of the study?
Minor editing of English language required.
Author Response
- REVIEWER 4 –
-
Comments and Suggestions for Authors
Based on the manuscript, it appears that the authors conducted a systematic review of clinical trials on the use of therapeutic vaccines for the treatment of high-grade cervical intraepithelial neoplasia (CIN) associated with human papillomavirus (HPV). The authors aimed to evaluate the safety, efficacy, and immunogenicity of therapeutic vaccines in this context.
The authors searched several databases for relevant studies and identified 16 that met their criteria. They found that the therapeutic vaccines were heterogeneous in terms of their formulation, dose, intervention protocol, and routes of administration, which made it unfeasible to perform a meta-analysis.
Overall, the text provides a clear and concise summary of the author's research and findings, highlighting the need for further research in this area. The text could be improved by providing more detailed information on the specific studies included in the systematic review, and by discussing potential limitations of the research.
Response: Thank you for your comments. We have added new limitations of this systematic review in accordance with you and also with requested by others reviewers
- Introduction Section:
HPV is a major virus associated with the development of cervical cancer in women, as mentioned in the first and second paragraphs. Considering the significance of this virus, it is crucial to explore effective treatment options. Currently, there are various therapeutic approaches available for this case, including immunotherapy, virotherapy, chemotherapy, and more. These treatments aim to combat HPV and its associated diseases, such as high-grade cervical intraepithelial neoplasia (CIN 2/3), and offer potential solutions for patients. By investigating and evaluating these therapies, we can contribute to improving the management and outcomes of HPV-related conditions.
Therefore, I recommended some various therapeutic to use and check according to your text:
1) DOI: 10.1016/j.micpath.2020.104438
2) DOI: 10.1186/s12985-021-01571-7
3) DOI: 10.1016/j.sjbs.2021.06.043
Response: Thank you for these suggestions. In this sense, we have added one paragraph in the discussion section addressing this subject and we have incorporated recommended references.
- Material Methods section:
1) Can you provide an overview of the inclusion and exclusion criteria used in the selection of studies for this systematic review? Specifically, what specific criteria were employed to determine the eligibility of studies in terms of study design, patient population, therapeutic vaccines evaluated, and the endpoints of safety, efficacy, and immunogenicity?
Response: All questions were already well detailed in the study protocol published by our group in BMJ Open (reference 29), and can also be found in the methods section.
2) Can you provide more information about the process of manually analyzing the references in the included studies to find additional relevant studies? Specifically, how was the manual analysis conducted, and what criteria were used to determine the relevance of the identified references?
Response: Two authors independently screened the entire reference list of 16 included studies to ensure that we did not miss any potentially eligible studies. We also retrieved previous systematic reviews on the subject and checked the included studies.
3) Based on the given text materials and Methods, here are some important questions that can be derived:
- What is the current understanding of the relationship between HPV and high-grade cervical intraepithelial neoplasia (CIN) as well as cervical cancer?
- What is the current status of research on therapeutic vaccines for the treatment of high-grade CIN associated with HPV?
Response: These questions are well addressed in the introduction and discussion of the manuscript.
- Were there any restrictions on the data or language in the studies included?
Response: No. This is stated in method line 159. There was no date or language restriction in the search strategy
- What were the primary endpoints assessed in the clinical trials?
- What were the key findings regarding the safety and efficacy of therapeutic vaccines?
- Were there any commonalities or differences observed among the therapeutic vaccines in terms of formulation, dose, intervention protocol, or routes of administration?
- Why was a meta-analysis deemed unfeasible for the collected data?
Response: In fact, we even conducted meta-analyses, but as the studies were very heterogeneous, the combination was most often two or two studies and the data were very sparse... The statistician of the group and the experts of review studies recommended not present the meta-analysis but a qualitative synthesis of the results.
- Was there any correlation observed between immunogenicity and clinical response?
- What are the implications of the findings for the development and future research of therapeutic vaccines?
- In conclusion, what are the main recommendations put forward by the authors of the study?
Response: All these questions are well addressed in the methods and discussion, concusion of the manuscript.
Comments on the Quality of English Language
Minor editing of English language required.
Response: English language was revised.
Reviewer 5 Report (New Reviewer)
See attached document

See comments in attached document
Author Response
- REVIEWER 5
Brief Summary:
The authors conducted an extensive literature review on progress made on the safety, efficacy, and immunogenicity of therapeutic vaccines for patients with high-grade cervical intraepithelial neoplasia 3 (CIN 2/3). The review is mostly descriptive as the vaccines administered and the methods used to evaluate them vary greatly. Therefore, it is impossible to draw any compelling conclusions re: what direction the further development of any of these vaccines should take. In my opinion, this paper can be viewed as a comprehensive literature review. The paper could benefit from an analysis as to why there was no follow-up development on these vaccine candidates which are being described having “favorable results”.
Response: This study is not a narrative review but a systematic review conducted with high methodological rigor. Unlike our review reported here, a narrative review does not present an advanced and qualified search strategy in the various databases, does not have well-established inclusion and exclusion criteria, the selected articles are impregnated with the subjectivities of the authors, as there are no criteria clear, there is no methodological assessment of the risk of bias of the included studies; etc.
Our study is a systematic review without meta-analysis conducted with high adherence to the PRISMA Statement 2020 checklist (Page et al., 2021), whose study protocol was published in the BMJ Open (Golçalves et al., 2019).
General Comments:
Overall, the readability of the paper can be improved. A couple of suggestions are provided under specific comments below.
Quite a few of studies included in this review were very small (9 out of 16 reviewed) had less than 25 subjects and did not include a placebo control group making it again very difficult to draw any meaningful conclusions. It would be important to understand how the inclusion of these studies benefits the paper overall.
The title of the paper is somewhat misleading as this concerns a qualitative analysis versus a systematic analysis of a specific intervention. There are too many variables among the vaccine candidates that are included in the analysis to draw any meaningful conclusions.
The review would benefit from better organizing the papers included in the analysis (which papers are related?). For example, it appears that Sheets et al 2003 reports initial findings and Garcia et al 2004 report on follow-up development of the same vaccine now including also a DNA encoding the E6 protein.
Presenting the data based on vaccine type (DNA, versus peptide, versus recombinant viral vector) might allow some general conclusions as to the most promising vaccine approach. Same comment relates to the route of administration and or the use of adjuvants Y/N.
The online databases were searched from date of inception through October 31, 2022. The authors need to ensure that no important publications are missed since then as it is August 2023 now.
Response: We have updated the Search in August 2023, but no new studies were added because they did not meet all previously established inclusion criteria.
The discussion includes a further review of additional many papers that begs the question: Why weren’t they included in the analysis in the first. For example, Reference 51 seems to clearly meet the inclusion criteria and concerns a large RCT study! The authors need to explain how they missed this (and other studies) in their analysis as it begs the question how thorough was the search that was conducted and how relevant were the exclusion criteria?
Response: These trials did not report the immunogenicity endpoint along with the other two endpoints (safety and efficacy). We would like to point out that our review had this criterion established from the beginning. Therefore, it is not possible to transpose as well as include it in the results section, as we would be violating the rigor of the method. In order to balance this issue raised, we have added this issue as a limitation of the review.
“Another limitation of the present systematic review is due to the fact that we used very rigorous selection criteria, i.e., included the three endpoints (safety, efficacy and immunogenicity) simultaneously, which made us exclude large and important trials in this field from the sample, which addressed one or two of the outcomes (although such trials were addressed in the discussion section, due to their great contribution to this area of knowledge).”
In addition, we recommend that future systematic reviews in this field take into account at least two of the outcomes reported here in order to expand the sample of potentially included studies for evaluation.
Specific Comments:
Title: Suggest changing title to include that this concerns a qualitative analysis.
Response: We completely disagree with this comment, as already explained above and in addition two authors are experts in the method of systematic review studies.
Simple summary:
Rephrase sentence Line 19 through 22 to improve readability
Response: OK. Done. Rephrased.
Line 22; Replace “Systematic” with “Qualitative”
Response: We cannot accept this suggestion, as explained above.
Line 25 -27: Suggest removing this sentence or explain “inconsistent results” and what the implications are.
Abstract:
Line 34: Suggest to replace “Analyzing” with “Reviewing”
Response: OK. Done. Thanks!
Line 45: Insert “In summary” before Therapeutic
Response: OK. Done. Thanks!
Line 46: Replace “for” with “to”
Response: OK. Done. Thanks!
Introduction:
Line 56: Replace “further” with “due to”
Response: OK. Done. Thanks!
Line 68: HPV uses -remove plural from HPVs
Response: OK. Done. Thanks!
Line 68 – 78: Clarify the role of the oncoproteins E6 and E7 and include appropriate references for example for Line 75 -77.
Line 80: Remove “Thus”
Response: OK. Done. Thanks!
Methods and Materials:
Results:
Figure 1. Suggest removing all analysis of studies identified via other methods from Figure 1 as none of these studies were included in the qualitative analysis. Add to line 193 that none of the 189 records from additional sources were included.
Response: OK. Done. Thanks!
Line 228: Remove “s” from oncoproteins
Response: OK. Done. Thanks!
Table 2: Under Garcia et al. Vaccine Type replace “e” with “&” sign
Response: OK. Done. Thanks!
Line 300: Correct “humoral”
Response: OK. Done. Thanks!
Figure 3: Replace Viral “Depuration” with Virus “Reduction”
Response: OK. Done. Thanks!
The inclusion criteria are Efficacy, Safety and Immunogenicity but in Figure 3 and the legend (Line 311-312) not all studies allow an assessment of these parameters – which begs the question why were the studies included?
Response: Sentence rephrased
Line 313: Replace “Once” with “Since”
Response: OK. Done. Thanks!
Discussion:
See overall assessment under general comments above. It seems like more advanced clinical studies were done that were not included in this analysis.
Response: These important clinical trials did not report the immunogenicity endpoint along with the other two endpoints (safety and efficacy). We would like to point out that our review had this criterion established from the beginning. Therefore, it is not possible to transpose as well as include it in the results section, as we would be violating the rigor of the method. In order to balance this issue raised, we have added this issue as a limitation of the review.
“Another limitation of the present systematic review is due to the fact that we used very rigorous selection criteria, i.e., included the three endpoints (safety, efficacy and immunogenicity) simultaneously, which made us exclude large and important trials in this field from the sample, which addressed one or two of the outcomes (although such trials were addressed in the discussion section, due to their great contribution to this area of knowledge).”
In addition, we recommend that future systematic reviews in this field take into account at least two of the outcomes reported here in order to expand the sample of potentially included studies for evaluation.
Round 2
Reviewer 1 Report (New Reviewer)
Overall the authors have been responsive to the comments and suggestions. Thank you.
There are still some areas that need editing, including ne of the new comments in line 192.
Author Response
Reviewer 1
Comments and Suggestions for Authors - Overall the authors have been responsive to the comments and suggestions. Thank you.
- Response: Thank you so much.
Comments on the Quality of English Language - There are still some areas that need editing, including one of the new comments in line 192.
- Response: ok. Done!
Reviewer 4 Report (New Reviewer)
Accepted for I checked the comments that were recommended in my previous review, and I am happy to inform you that the authors have addressed all of them diligently. Therefore, I believe it is now suitable for publishing in the current journal.
Dear Editor,
Greetings,
I checked the comments that were recommended in my previous review, and I am happy to inform you that the authors have addressed all of them diligently. Therefore, I believe it is now suitable for publishing in the current journal.
Best regards,
AS
Author Response
Reviewer 4
Comments and Suggestions for Authors
Accepted for I checked the comments that were recommended in my previous review, and I am happy to inform you that the authors have addressed all of them diligently. Therefore, I believe it is now suitable for publishing in the current journal. Comments on the Quality of English Language Dear Editor, Greetings, I checked the comments that were recommended in my previous review, and I am happy to inform you that the authors have addressed all of them diligently. Therefore, I believe it is now suitable for publishing in the current journal Best regards, AS
- Response: Thank you so much.
Reviewer 5 Report (New Reviewer)
The authors did not adequately address my review comments.

Author Response
Reviewer 5
The authors did not adequately address my review comments. peer-review-31865896.v1.pdf
Brief Summary: The authors conducted an extensive literature review on progress made on the safety, efficacy, and immunogenicity of therapeutic vaccines for patients with high-grade cervical intraepithelial neoplasia 3 (CIN 2/3). The review is mostly descriptive as the vaccines administered and the methods used to evaluate them vary greatly. Therefore, it is impossible to draw any compelling conclusions re: what direction the further development of any of these vaccines should take. In my opinion, this paper can be viewed as a comprehensive literature review. The paper could benefit from an analysis as to why there was no follow-up development on these vaccine candidates which are being described having “favorable results”.
Response: This study is not a narrative review but a systematic review conducted with high methodological rigor. Unlike our review reported here, a narrative review does not present an advanced and qualified search strategy in the various databases, does not have well-established inclusion and exclusion criteria, the selected articles are impregnated with the subjectivities of the authors, as there are no criteria clear, there is no methodological assessment of the risk of bias of the included studies; etc. Our study is a systematic review without meta-analysis conducted with high adherence to the PRISMA Statement 2020 checklist (Page et al., 2021), whose study protocol was published in the BMJ Open (Golçalves et al., 2019).
General Comments: Overall, the readability of the paper can be improved. A couple of suggestions are provided under specific comments below. Quite a few of studies included in this review were very small (9 out of 16 reviewed) had less than 25 subjects and did not include a placebo control group making it again very difficult to draw any meaningful conclusions. It would be important to understand how the inclusion of these studies benefits the paper overall. The title of the paper is somewhat misleading as this concerns a qualitative analysis versus a systematic analysis of a specific intervention. There are too many variables among the vaccine candidates that are included in the analysis to draw any meaningful conclusions. The review would benefit from better organizing the papers included in the analysis (which papers are related?). For example, it appears that Sheets et al 2003 reports initial findings and Garcia et al 2004 report on follow-up development of the same vaccine now including also a DNA encoding the E6 protein. Presenting the data based on vaccine type (DNA, versus peptide, versus recombinant viral vector) might allow some general conclusions as to the most promising vaccine approach. Same comment relates to the route of administration and or the use of adjuvants Y/N. The online databases were searched from date of inception through October 31, 2022. The authors need to ensure that no important publications are missed since then as it is August 2023 now.
Response: We have updated the Search in August 2023, but no new studies were added because they did not meet all previously established inclusion criteria.
The discussion includes a further review of additional many papers that begs the question: Why weren’t they included in the analysis in the first. For example, Reference 51 seems to clearly meet the inclusion criteria and concerns a large RCT study! The authors need to explain how they missed this (and other studies) in their analysis as it begs the question how thorough was the search that was conducted and how relevant were the exclusion criteria?
Response: These trials did not report the immunogenicity endpoint along with the other two endpoints (safety and efficacy). We would like to point out that our review had this criterion established from the beginning. Therefore, it is not possible to transpose as well as include it in the results section, as we would be violating the rigor of the method. In order to balance this issue raised, we have added this issue as a limitation of the review. “Another limitation of the present systematic review is due to the fact that we used very rigorous selection criteria, i.e., included the three endpoints (safety, efficacy and immunogenicity) simultaneously, which made us exclude large and important trials in this field from the sample, which addressed one or two of the outcomes (although such trials were addressed in the discussion section, due to their great contribution to this area of knowledge).” In addition, we recommend that future systematic reviews in this field take into account at least two of the outcomes reported here in order to expand the sample of potentially included studies for evaluation.
Specific Comments: Title: Suggest changing title to include that this concerns a qualitative analysis.
Response: We completely disagree with this comment, as already explained above and in addition two authors are experts in the method of systematic review studies.
Simple summary: Rephrase sentence Line 19 through 22 to improve readability
Response: OK. Done. Rephrased.
Line 22; Replace “Systematic” with “Qualitative”
Response: We cannot accept this suggestion, as explained above.
Line 25 -27: Suggest removing this sentence or explain “inconsistent results” and what the implications are. Abstract: Line 34: Suggest to replace “Analyzing” with “Reviewing”
Response: OK. Done. Thanks!
Line 45: Insert “In summary” before Therapeutic
Response: OK. Done. Thanks!
Line 46: Replace “for” with “to”
Response: OK. Done. Thanks!
Introduction: Line 56: Replace “further” with “due to”
Response: OK. Done. Thanks!
Line 68: HPV uses -remove plural from HPVs
Response: OK. Done. Thanks!
Line 68 – 78: Clarify the role of the oncoproteins E6 and E7 and include appropriate references for example for Line 75 -77. Line 80: Remove “Thus”
Response: OK. Done. Thanks!
Methods and Materials: Results: Figure 1. Suggest removing all analysis of studies identified via other methods from Figure 1 as none of these studies were included in the qualitative analysis. Add to line 193 that none of the 189 records from additional sources were included.
Response: We added this information, however, we cannot remove it from the Figure, as it is part of the method. OK. Done. Thanks!
Line 228: Remove “s” from oncoproteins
Response: OK. Done. Thanks!
Table 2: Under Garcia et al. Vaccine Type replace “e” with “&” sign
Response: OK. Done. Thanks!
Line 300: Correct “humoral”
Response: OK. Done. Thanks!
Figure 3: Replace Viral “Depuration” with Virus “Reduction”
Response: OK. Done. Thanks!
The inclusion criteria are Efficacy, Safety and Immunogenicity but in Figure 3 and the legend (Line 311-312) not all studies allow an assessment of these parameters – which begs the question why were the studies included?
Response: Sentence rephrased
Line 313: Replace “Once” with “Since”
Response: OK. Done. Thanks!
Discussion: See overall assessment under general comments above. It seems like more advanced clinical studies were done that were not included in this analysis.
Response: These important clinical trials did not report the immunogenicity endpoint along with the other two endpoints (safety and efficacy). We would like to point out that our review had this criterion established from the beginning. Therefore, it is not possible to transpose as well as include it in the results section, as we would be violating the rigor of the method. In order to balance this issue raised, we have added this issue as a limitation of the review. “Another limitation of the present systematic review is due to the fact that we used very rigorous selection criteria, i.e., included the three endpoints (safety, efficacy and immunogenicity) simultaneously, which made us exclude large and important trials in this field from the sample, which addressed one or two of the outcomes (although such trials were addressed in the discussion section, due to their great contribution to this area of knowledge).” In addition, we recommend that future systematic reviews in this field take into account at least two of the outcomes reported here in order to expand the sample of potentially included studies for evaluation.
This manuscript is a resubmission of an earlier submission. The following is a list of the peer review reports and author responses from that submission.
Round 1
Reviewer 1 Report
Well researched and written study. Exciting.